# Effectiveness of spatially targeted interventions for control of HIV, tuberculosis, leprosy and malaria: a systematic review

McEwen Khundi [1,2] James R Carpenter [2,3] Marriott Nliwasa [4]
Ted Cohen [5] Elizabeth L Corbett [1,6] Peter MacPherson [1,2,7]

For numbered affiliations see end of article.

**Correspondence to**
McEwen Khundi;
mcewenkhundi@gmail.com

## ABSTRACT

**Background** As infectious diseases approach global elimination targets, spatial targeting is increasingly important to identify community hotspots of transmission and effectively target interventions. We aimed to synthesise relevant evidence to define best practice approaches and identify policy and research gaps.

**Objective** To systematically appraise evidence for the effectiveness of spatially targeted community public health interventions for HIV, tuberculosis (TB), leprosy and malaria.

**Design** Systematic review.

**Data sources** We searched Medline, Embase, Global Health, Web of Science and Cochrane Database of Systematic Reviews between 1 January 1993 and 22 March 2021.

**Study selection** The studies had to include HIV or TB or leprosy or malaria and spatial hotspot definition, and community interventions.

**Data extraction and synthesis** A data extraction tool was used. For each study, we summarised approaches to identifying hotpots, intervention design and effectiveness of the intervention.

**Results** Ten studies, including one cluster randomised trial and nine with alternative designs (before–after, comparator area), satisfied our inclusion criteria. Spatially targeted interventions for HIV (one USA study), TB (three USA) and leprosy (two Brazil, one Federated States of Micronesia) each used household location and disease density to define hotspots followed by community-based screening. Malaria studies (one each from India, Indonesia and Kenya) used household location and disease density for hotspot identification followed by complex interventions typically combining community screening, larviciding of stagnant water bodies, indoor residual spraying and mass drug administration. Evidence of effect was mixed.

**Conclusions** Studies investigating spatially targeted interventions were few in number, and mostly underpowered or otherwise limited methodologically, affecting interpretation of intervention impact. Applying advanced epidemiological methodologies supporting more robust hotspot identification and larger or more intensive interventions would strengthen the evidence-base for this increasingly important approach.

**PROSPERO registration number** CRD42019130133.

## Strengths and limitations of this study

► This was a thoroughly conducted systematic review which only included published literature.

► We developed the search strategy with input from infectious disease experts, statisticians and librarians, and we only included studies that met our objective assessment criteria.

► We acknowledge that some studies might still have been missed even with the systematic approach that we used.

► The effect of publication bias cannot be ruled out as it is possible that studies that had negative results might not have been published; thereby, they would not have been captured by our search strategy.

► The studies had different interventions and outcomes as such a meta-analysis was not done contrary to the initial plans.

## INTRODUCTION

The world has made tremendous strides in controlling HIV, tuberculosis (TB), leprosy and malaria epidemics. Despite impressive achievements in reductions of new cases, considerable efforts are required to meet and maintain elimination targets.[1–4] Much of the current progress in disease control is attributable to the combination of facility based routine health services, supplemented by community based interventions.[1–4]

The success of disease control strategies provided through facility-based services relies on prompt recognition of symptoms by the patient, early health seeking by the patient, and correct recognition and management by health providers to provide early diagnosis and effective medical care.[1–4] However, health-seeking delays can be prolonged, and this strategy will also have no effect on transmission during subclinical illness.[2 5–7] In low-income settings, lack of faith in the quality of services, high opportunity and indirect costs

and non-adherence to syndromic management protocols can combine to cause substantial health-seeking and diagnostic delays, undermining the effectiveness of facility-based strategies based on early diagnosis and treatment.[2 4 5 8–12] Community interventions can be complementary to facility-based services but are more resource intensive, logistically challenging and are generally only justifiable when targeted at groups of people at risk or in a geographically defined area with high prevalence or incidence of disease.[1–4]

In communities, HIV, TB, leprosy and malaria cluster geographically into hotspots.[13–19] Hotspots are defined as areas of high incidence or prevalence compared with neighbouring geographical areas.[20] For malaria, which is vector-borne, environmental characteristics conducive to replication of anopheles mosquitos are a key consideration.[18] For HIV, TB and leprosy, diseases that are transmitted exclusively person-to-person, hotspots tend to be characterised by poverty, poor access to health services, overcrowding, concentrations of migrant populations, and poor housing.[13–18] Since hotspots are likely to be areas of relatively high transmission, targeting interventions at hotspots may prevent many more infections and cases of disease than similar efforts in low-transmission settings or untargeted efforts (online supplemental figure 1).[21] Public health interventions amenable to spatially defined hotspot targeting include: screening, case-finding, prevention (including vaccination), and improvement in access to services for diagnosis and treatment.[15 20]

Community health and prevention interventions frequently target people in slums or informal urban settlements.[22] In low and middle income countries, many millions of people live in informal urban settlements,[23] and disease control programmes would benefit from more precise identification of high priority areas within these to allow interventions to be delivered to communities where the impact is likely to be greatest.[24 25] At the moment targeting tends to use relatively crude epidemiological criteria such as age and gender.[1–4 26] But since cases of the diseases in this review are known to cluster geographically, targeting of all people in a carefully identified geographically defined hotspots offers an alternative criteria to define at-risk groups for interventions.[13–19]

The main aim of this systematic review was to systematically appraise evidence for spatially targeted community public health interventions directed towards major infectious diseases that are transmitted in different ways: HIV is predominantly sexually transmitted; TB by respiratory droplet transmission; leprosy by direct contact and droplet transmission; and malaria by vector-borne transmission. We aimed to summarise lessons learnt from evaluation of spatially targeted interventions against these diseases and to make recommendations for researchers, policymakers and disease control programmes, as well as to inform the design and evaluation of future studies investigating spatially targeted interventions.

## METHODS
### Study design
Systematic literature review.

### Public involvement statement
We developed the research question for this project after realising the evidence gap on the effectiveness of spatially targeted interventions despite the growing interest that this approach has. To develop the scope of the research question we engaged experts in public health interventions of HIV, TB, leprosy and malaria. The search strategy was also developed in consultation with the experts. The results of the research will be disseminated to researchers and policy-makers at local and international conferences.

### Search strategy
We systematically searched the literature using major subject headings and keywords to identify published studies meeting the inclusion criteria following our published protocol (PROSPERO ID: CRD42019130133). Databases searched included: Medline, Embase, Global Health, Web of Science and the Cochrane Database of Systematic Reviews. The three central concepts included in our search strategy (online supplemental table 1) were the disease condition (HIV, TB, leprosy and malaria); space (techniques used to identify hotspots); and community interventions.

### Eligibility criteria
We included studies published between the period from 1 January 1993 and 22 July 2019 and then updated to 22 March 2021. The start of the search period was chosen as the year that TB was declared as a 'global emergency' by the WHO. We focused on studies of spatially targeted interventions for HIV, TB, leprosy and malaria with the aim of identifying all the available literature on the effectiveness of spatially targeted intervention in the selected diseases, and then summarising the findings. We did not limit inclusion by age group of participants or geographical region. We included randomised controlled trials, non-randomised observational studies, before-and-after studies and time-series analysis studies. The following articles were excluded: editorials; narratives; systematic reviews; case studies; case reports; case-control studies; contact investigation studies; and non- spatially targeted community studies.

### Selection of studies and data extraction
We imported studies into an Endnote (Thompsons Reuters) database, and duplicates were removed. MK and MN independently screened the title and abstracts against inclusion and exclusion criteria (online supplemental table 2), identified studies eligible for full-text review, and subsequently reviewed the full text of each selected study independently. Discrepancies between MK and MN were resolved with discussion, and where agreement could not be reached, two other reviewers (PMP and JRC) participated in a consensus review.

We developed a data extraction form that was piloted on a sample of selected manuscripts. This form was independently completed for each study selected for full-text review by MK and MN.

## Assessment of study quality

Two reviewers (MK and MN) assessed the quality of each study. We used separate tools to assess the methodological quality of cluster randomised trials (CRTs) (Cochrane Collaboration risk of bias tool)[27] and non-randomised studies (ROBINS-I: a tool for assessing risk of bias in non-randomised studies of interventions).[28]

## Definitions

We defined spatially targeted interventions as community interventions that targeted hotspots of disease. Hotspots were defined as subdistrict geospatially defined areas that had a high number of incident or prevalent cases of the infection or disease compared with surrounding areas, studies that based their hotspots as around diagnosed index cases were not included. In practice, as definitions varied considerably between studies, we extracted and compared hotspot definitions between studies.

## Statistical analysis

The main objective of the systematic review was to compare the impact of the intervention in the intervention hotspot areas compared with control areas. We report the outcomes defined by the included studies, with outcomes being prevalence, incidence, case notification rates and number needed to screen to identify a positive case. We calculated measures of effect and uncertainty in effect estimates using data available in manuscripts where these were not provided by the authors using R V.3.61 (R Core Team). Because of high anticipated heterogeneity between and within diseases and interventions investigated, we decided not to undertake meta-analysis.

## Ethics statement

This review used published data, and ethical review was not required.

## Data sharing statement

The search strategy and summary data tables for this systematic review have all been included in the main text and in the appendix section.

## RESULTS

A total of 3919 unique abstracts were identified by the search strategy, from which 3886 were excluded from the title and abstract review, leaving 28 studies that were reviewed as full-text articles. Overall, 10 studies met the inclusion criteria (online supplemental figure 2). The reasons for exclusion are found in online supplemental table 2.

One USA-based study was identified for HIV. Three TB studies were found: all three studies were from the USA. Three leprosy studies were found: two from Brazil and

one study from the Federated States of Micronesia. For malaria, we identified three studies: one each from India, Indonesia and Kenya. The most common study design was implementation demonstration studies (ie, studies in which spatially targeted interventions were introduced without random allocation of hotspots to study arms). Nine studies had this design: one HIV, three TB, three leprosy and two malaria. We identified one CRT, which evaluated interventions targeted against malaria (online supplemental table 3).

Table 1 summarises the characteristics of the included studies, presenting outcomes, measures of intervention effects and study quality/risk of bias assessment results. Table 2 synthesises methods used to geolocate cases and identify hotspots in the included studies. [10] Online supplemental table 3 has details on why studies were excluded following full text review.

## Spatially targeted HIV interventions

In Goswami et al,[29] cases of TB (N=150), HIV (N=665) and syphilis (N=155) notified between 1 January 2005 and 31 December 2007 in Wake County, North Carolina, USA, were geocoded to households. Kernel density maps of the three diseases were generated; maps identified two areas with the densest number of cases (hotspots). Hotspots were defined as areas with more than 10 cases of either TB, HIV or syphilis notified per square mile during the 3 years. A map of streets and local businesses of Wake County was used to identify the locations of the hotspot areas.

Between 2 June 2009 and 3 November 2011, adult community screening for HIV was done in the defined hotspots at specific sites by community nurses and disease intervention specialists from the HIV, syphilis and TB clinics at the county health department. Of 247 community participants screened by the study, 240 had valid HIV test results. Prevalence of HIV in hotspot areas was compared with that among patients presenting to a sexually transmitted disease (STD) clinic located outside of the hotspot areas. HIV prevalence was higher among community screened participants (8/240, 3%, 95% CI 1.4% to 6.5%) compared with the Wake County STD clinic (64/15 936, 0.4%, 95% CI 0.3% to 0.5%) with a risk ratio of 8.3 (95% CI 4.0 to 17.1, p<0.001). Community HIV screening identified eight HIV positive cases, only one of whom was previously undiagnosed.

## Spatially targeted TB interventions

In Moonan et al,[24] notified TB cases from 1 January 1993 to 31 December 2000 (N=991) were geolocated to zip codes in Tarrant County, north-central Texas, USA. Three zip codes were found to have the highest TB notification rates per 100 000 of the population, in addition to having genotypically clustered *Mycobacterium tuberculosis* isolates.[24] The TB notification rates of the zip codes were 94, 55 and 32 cases per 100 000 while the case notification rate for the whole county for the same period was 5.9 cases per 100 000; 95 of 117 (81%) isolates were genotypically

**Table 1** Characteristics of spatially targeted interventions for HIV, TB, leprosy and malaria

| Reference, year | Outcome | Effect of intervention* | Risk of bias assessment |
|---|---|---|---|
| **HIV** | | | |
| Goswami et al[29] | Comparison of case detection yield between hotspot intervention areas and a county STD clinic over the same period (2009–2011) | HIV prevalence was higher among community screened participants (8/240, 3%, 95% CI 1.4% to 6.5%) compared with the Wake County STD clinic (64/15936, 0.4%, 95% CI 0.3% to 0.5%) with a risk ratio of (8.3, 95% CI 4.0 to 17.1), p<0.001. | Moderate |
| **Tuberculosis (TB)** | | | |
| Moonan et al[24] | Yield of TB case detection in hotspots. | Targeted screening identified one person with TB for every 83 screened and one person with LTBI for every five screened. The yield of the targeted approach was considered to be more than what would be expected in a county with an active TB notification rate of 5.7 per 100000 population year. | Critical |
| Goswami et al[29] | Comparison of case detection yield of LTBI between hotspot areas and county TB clinic over the same period (2009–2011) | LTBI prevalence was higher among community screened participants (36/234, 15%, 95% CI 11.0% to 21.7%) versus (541/9024, 6%, 95% CI 5.6% to 6.6%) at the TB clinic with a risk ratio of (2.5, 95% CI 1.9 to 3.5), p<0.001. | Moderate |
| Cegielski et al[31] | A before and after intervention comparison of mapped TB notification rates between 1985–1995 and 1996–2006 | TB notification rates in the targeted hotspots declined from 39.6 per 1 000 000 people per year (95% CI 30.4 to 48.8) from 1985 to 1995 to zero from 1996 to 2006 (p<0.001) | Serious |
| **Leprosy (three studies)** | | | |
| De Souza Dias et al[32] | Percent of notified cases attributable to the intervention, and before and after intervention case notification rate comparison. | Active case finding identified 50% of the total cases that were diagnosed in 2005. The case notification rate in 2005 was higher compared with pre intervention year 2004, 9.34 per 10000 versus 5.16 per 10000, respectively. | Serious |
| Jim et al[33] | Yield of leprosy case detection during intervention period compared with the preintervention period and reduction in households that needed to be screened. | Eight-fold decrease in the number of households that needed to be screened from 2007 to 2009. While still identifying a similar number of new cases to prespatially targeted active case finding period 2002–2006. | Moderate |
| Barreto et al[25] | The yield of case detection of leprosy cases in school children in hotspot intervention schools versus in children from randomly selected schools. | In the hotspot school's (11/134, 8.2%, 95% CI 3.5% to 13.0%) students with a mean age of 10 years were diagnosed with leprosy. While (63/1592, 3.9%, 95% CI 3.0% to 4.9%) students from randomly selected schools with a mean age of 12 years were diagnosed with leprosy with a risk ratio of (2.1, 95% CI 1.1 to 3.8), p<0.05. | Moderate |
| **Malaria (four studies)** | | | |
| Srivastava et al[34] | Difference in absolute numbers of notified malaria cases between 2006 and 2007. | An absolute reduction in numbers of notified cases in 2007 (N=90829) from the notified cases in 2006 (N=96042), (5.7%, 95% CI 4.7% to 6.7%), p<0.05. | Critical |
| Herdiana et al[35] | Change of malaria notification rates from preintervention to postintervention period. | 30-fold reduction in malaria notifications from 3.83 per 1000 in 2008 to 0.13 per 1000 in 2011. | Serious |
| Bousema et al[36] | Change in parasite prevalence in the evaluation zones (1–500 m from hotspots) of intervention clusters versus control clusters | The first evaluation zone 1–249 m at eighth week (3.6%, 95% CI −2.6% to 9.7%), p=0.216and the second evaluation zone 250–500 m at eighth week (3.8%, 95% CI −2.4% to 10.0%), p=0.187. The first evaluation zone 1–249 m at 16th week (1.0%, 95% CI −7.0% to 9.1%), p=0.713and the second evaluation zone 250–500 m at 16th week (1.0%, 95% CI −8.3% to 10.4%), p=0.809 | Low |

*Results calculated from the data published in the papers.
LTBI, latent tuberculosis Infection; m, metres; N, number; STD, sexually transmitted disease.

**Table 2** Geolocation of cases and hotspot identification

| Reference, year | Geolocation of cases | Hotspot identification |
|---|---|---|
| **HIV (one study)** | | |
| Goswami et al[29] | Notified cases of either tuberculosis (TB) (N=150), HIV (N=665) or syphilis (155) between 1 January 2005 and 31 December 2007 were geolocated to households. The method for geolocation of the cases was not described in the paper. | A kernel density map of the cases was produced. Areas with the highest densities of three diseases of HIV, syphilis and TB (greater than 10 cases per square mile) were classified as hotspots. Two hotspot neighbourhoods were identified in the county. |
| **TB (four studies)** | | |
| Moonan et al[24] | Notified TB cases (N=991) from 1 January 1993 to 31 December 2000 in Tarrant County, north central Texas, USA were geolocated to zip codes using residential addresses and zip codes that patients gave at the time of diagnosis with the aid of a GIS software. | Areas with the highest TB notification rates and high percentage of genotypically clustered TB isolates were identified as hotspots. Three neighbourhood hotspots were identified. |
| Goswami et al[29] | Notified cases of TB (N=150), HIV (N=665) and syphilis (N=155) that were notified between 1 January 2005 and 31 December 2007 were geolocated to households. The method for geolocation of the households was not described in the paper | A kernel density map was developed. Areas with the highest densities of three diseases of HIV, syphilis and TB (greater than 10 cases per square mile) were classified as hotspots. Two hotspot neighbourhoods were identified in the county. |
| Cegielski et al[31] | Notified TB cases between 1985 to 1995 (N=128) and all notified LTBI from 1993 to 1995 (N=311) were geocoded to their households using the addresses that patients gave at the time of diagnosis. In addition, field workers tracked addresses to households to get household coordinates of addresses that failed to geolocate. | The points of cases were plotted on a map and areas with the densest clusters of points of cases were identified as hotspots, two neighbourhoods were identified in the county. |
| **Leprosy (three studies)** | | |
| De Souza Dias et al[32] | Notified leprosy cases that occurred between 1998 and 2002 (N=368) in the municipality of geocoded to households. The method for geolocation of cases was not described. | Density map with a radius of 100 m of the notified leprosy cases was produced. Four hotspot areas were identified |
| Jim et al[33] | Notified leprosy cases from 2002 to 2006 (N=502) were geolocated to households. Field workers visited all notified cases to get household GPS coordinates using a GIS device. | A density map based on 1 mile radius of the notified leprosy cases. Areas with high concentration of cases classified as hotspots. |
| Barreto et al[25] | Notified leprosy cases from January 2004 to February 2010 (n=633) were geocoded to households. Field workers visited households of registered cases to collect GPS coordinates. | Hotspots were identified using the Kulldorff's spatial scan statistic and by stratification of the leprosy notified rates. Two hotspots were identified. |
| **Malaria (four studies)** | | |
| Srivastava et al[34] | Notified malaria cases between 2000 and 2005 were obtained from the State Department of Health based on the cases notified in clinics in the blocks or districts. | Blocks or districts with a percentage of Plasmodium falciparum malaria cases of all notified cases that was either 100% or consistently greater than 30% from 2000 to 2005 or greater than 70% in 2005 |
| Herdiana et al[35] | Notified malaria cases from 2007 to 2008 in addition to other self-reported malaria cases that were found during a survey (n=319) were geocoded to households. Field workers obtained the GPS coordinates of households using GIS devices. | Villages that had the majority of malaria cases were classified as hotspots. 14 out of 18 villages were identified |
| Bousema et al[36] | June and July 2011, 17 503 individuals tested in a malaria prevalence survey for the prevalence of P. falciparum antibodies (AMA-1 or MSP-10). Field workers collected the GPS coordinates of the households using GPS devices. | Segments of the study area were scanned in the 2×4 km rolling windows and areas with higher (p<0.05) prevalence of antibodies and age-adjusted antibody density than the local average values were identified as hotspots. |

GIS, geographical information system; GPS, global positioning system; m, metres; N, number; STD, sexually transmitted disease.

clustered in these three zip codes.[30] Between 1 September 2002 and 31 December 2004 community-based organisations offered screening for TB and latent TB infection (LTBI) to hotspot residents. Overall, 3645 individuals were screened. 1.2% (N=44) people were diagnosed and treated for active TB, and 18.6% (N=681) were diagnosed and treated for LTBI. This targeted screening identified one person with active TB for every 83 screened, and one person with LTBI for every five screened. The yield of the targeted approach was considered to be more than what would be expected in a county with an active TB notification rate of 5.7 per 100 000 population year.

Goswami et al[29] (see above in HIV section) also offered screening for LTBI to the same individuals who were offered HIV testing. Both tests were performed from the same blood sample. Hotspot prevalence was compared with LTBI prevalence at the Wake County TB clinic, which was outside of the hotspot areas. Latent TB testing at the TB clinic was offered to high risk individuals who had close contact with a recently diagnosed active TB cases, refugees and those referred by primary care providers or employers. 234/247 had valid LTBI screening results. LTBI prevalence was higher among community screened participants (36/234, 15%, 95% CI 11.0% to 21.7%) versus 6% (95% CI 5.6% to 6.6%) at the TB clinic with a risk ratio of 2.5 (95% CI 1.9 to 3.5, p<0.001).

In Cegielski et al,[31] notified TB cases in Smith County, Texas USA between 1985 to 1995 (N=128) and all notified LTBI cases from 1993 to 1995 (N=311) were geocoded to their households. The geocoded cases were loaded into geographical information systems software to produce a point map of both active TB and LTBI cases. The two densest neighbourhoods of TB and LTBI cases were identified visually from the map. In 1996, study field workers went door to door in these neighbourhoods and offered tuberculin skin testing (TST). Of 2258, 1236 had LTBI testing and received results, 229/1236 (18.5%) were TST positive and 147 received treatment. To assess the intervention, the notified TB cases from 1996 to 2006 in Smith County were mapped to do a before and after intervention comparison. The TB notification rates in the targeted hotspots declined from 39.6 per 1 000 000 people per year (95% CI 30.4 to 48.8) from 1985 to 1995 to zero from 1996 to 2006 (p<0.001). While for the entire Smith County the TB notification rates reduced from 8.1 per 1 000 000 people per year (95% CI 5.2 to 11.0) from 1985 to 1995 to 3.7 per 100 000 people per year (95% CI 1.2 to 6.1) from 1996 to 2006 (p<0.001).

### Spatially targeted leprosy interventions

In De Souza Dias et al,[32] leprosy cases notified between 1998 and 2002 (N=368) in the municipality of Mossoro, Rio Grande do Norte in Brazil were geocoded to households. The geocoded cases were used to create a density map with a radius of 100 m, and four neighbourhoods with the highest concentration of cases were identified as hotspots. Four active case finding (ACF) campaigns were conducted in these hotspots between March and September 2005. Study team members went door to door to identify people with symptoms of leprosy and referred them to the nearest primary health clinic. Five hundred twelve possible leprosy cases were referred, and 104 leprosy cases were diagnosed. The cases identified through hotspot ACF represented 50% of the total cases diagnosed in the city in 2005. In addition, the case notification rate in 2005 was higher than in 2004; 9.34 per 10 000 versus 5.16 per 10 000, respectively.

In Jim et al,[33] in the state of Pohnpei in the Federated States of Micronesia, notified leprosy cases from 2002 to 2006 (N=502) were geocoded to households, producing a point-density map of 1 mile radius; areas with the densest areas were identified for ACF. During 2007 to 2008, ACF teams undertook door to door visits and screened household members for leprosy. There was an eightfold statewide decrease in the number of households that were screened between 2007 and 2009, while the number of identified cases was similar to the prespatially targeted ACF campaign period of 2002–2006.

In Barreto et al,[25] notified leprosy cases from the two municipalities of Castanhal and Oriximinal in the state of Para, Brazil from January 2004 to February 2010 (N=633) were geocoded to households. The notified leprosy cases were aggregated by census tract to calculate census case notification rates, and a spatially empirical case detection rate was used to smooth notification rates. Census tracts were classified into four categories, with the highest category assigned to areas with >40 cases per 100 000 population. Kulldorff's spatial scan statistic was used to define groups of hotspot census tracts with statistically higher than average notification rates. Two schools were selected: one within a hotspot and one from a census tract with ≥40 notified leprosy cases per 100 000 populations. At these schools, 134 students with a mean age of 10 years were screened and 11/134, 8.2% (95% CI 3.5% to 13.0%) were diagnosed with leprosy based on clinical signs and symptoms. The diagnostic yield in hotspot area schools was significantly higher than in students from randomly selected schools from eight municipalities in the state of Para in a cross-sectional study between 2009 to 2011: 63/1592, 3.9% (95% CI 3.0% to 4.9%) with a risk ratio of 2.1 (95% CI 1.1 to 3.8).

### Malaria

In Srivastava et al,[34] notified malaria cases between 2000 and 2005 obtained from the State Department of Health in Madhya Predesh were aggregated at block and district level. Malaria hotspots were defined based on the percentage of malaria cases that had *Plasmodium falciparum* malaria of all notified cases from 2000 to 2005. Unspecified targeted malaria interventions in 2007 were evaluated for effectiveness by comparing overall number of confirmed cases notified in 2006 with those notified in 2007. Absolute numbers of notified cases decreased by 5.7% (95% CI 4.7% to 6.7%) from 96 042 in 2006 (p<0.05).

In Herdiana *et al*,[35] in Sabang island, Indonesia, documented malaria cases from 2007 to 2008 and self-reported malaria cases identified during a survey (N=319), were geocoded to households to produce point maps that classified 14 out of 18 villages as hotspots, based on absolute number of cases. From May 2010, home visits were conducted twice a month in hotspot villages, and only once in non-hotspot villages. Malaria blood smears were taken from anyone with current or recent history of fever, with smear-positive participants referred for treatment. Household contacts and neighbours within 500 m of smear-positive participants were also screened. Interventions in hotspot areas included: improving malaria diagnostic labs, introduction of Artemisinin-based Combination Therapy for malaria treatment, scale-up of indoor residual spraying (IRS) and distribution of long-lasting insecticide-treated nets (LLINs). The incidence of malaria in hotspot areas decreased by 30-fold from 3.18 to 0.13 per 1000 population from 2008 to 2011.[34–36]

In Bousema *et al*,[36] between June and July 2011 in Rachuonyo, western Kenya, 17503 individuals were screened for a malaria prevalence survey, including *P. falciparum* antibodies. SaTScan software was used to define hotspots based on the prevalence of antibody-positivity and age-adjusted antibody density. In total 27 hotspots were identified. A randomised controlled trial randomly allocated ten hotspots 1:1 to either intervention or control. The intervention activities were weekly larviciding of stagnant water bodies, provision of LLINs, IRS and mass drug administration (MDA). MDA was only administered to households that had a confirmed malaria case; febrile individuals (temperature >37.5°C) and children aged 6 months–15 years were offered malaria rapid diagnostic tests. The following standard interventions were available to both arms, hotspots and evaluation zones (the area 500 m around each hotspot): annual IRS, case management at health facilities and distribution of LLINs from antenatal clinics.

There was no statistically significant difference in parasite prevalence in evaluation zones (area around the hotspot) at 8 weeks and 16 weeks postintervention time points. The first evaluation zone (1–249 m) at the eighth week found 3.6% (95% CI −2.6% to 9.7%, p=0.216) and the second evaluation zone (250–500 m) at the eighth week 3.8% (95% CI −2.4% to 10.0%, p=0.187). Neither was there any significant difference in the first evaluation zone (1–249 m) at the 16th week: 1.0% (95% CI −7.0% to 9.1%, p=0.713) and the second evaluation zone (250–500 m) at 16th week: 1.0% (95% CI −8.3% to 10.4%, p=0.809).

### Influence of study quality on results

The risk of bias assessment of the studies in this review focused on how the hotspots were selected and how the intervention were assessed in each of the studies that were included. The studies in this review were found to have issues that would make them susceptible to risk of bias. Hotspot selection was based mainly on notified cases.[24 25 29 32–35 37 38] Relying on notified cases alone can introduce selection bias because notified cases can over-represent cases from areas that have good access to health systems.[15]

Further, identified geographical hotspots need to be investigated for stability overtime to ensure that they display spatio-temporal consistency in notification trends.[15 39] This was not accounted for in the studies included in this review.[24 25 29 32–38 40]

The majority of the identified studies were implementation/pragmatic studies.[24 25 29 31–35 40 41] These did not randomise hotspots to interventions and therefore the studies might have compared outcomes of the study in groups of people that had baseline characteristics that were different.

## DISCUSSION
### Main findings

The key finding of this review is that, despite increasing enthusiasm for the application of spatially targeted interventions in public health research addressing infectious diseases,[42] very few studies have rigorously evaluated the effectiveness of such approaches against non-hotspot comparator areas. We identified only 10 studies conducted since 1993, nearly all of which had substantial limitations in how hotspots were defined and identified, how spatially targeted interventions were evaluated, or how comparator areas were selected. With the limited evidence available, we found some suggestion that hotspot-targeted interventions for TB, malaria and leprosy may be efficient and effective approaches in increasing diagnostic yield, reducing unnecessary screening, and perhaps in improving disease epidemiology. However, almost all the studies were vulnerable to bias due to regression to the mean: hotspots identified by highest prevalence/incidence will typically see prevalence/incidence decline, even in the absence of an intervention. As such, spatially targeted approaches hold promise, but require further evaluation in high-quality studies to guide policy-makers considering implementing this approach.

### Strengths and limitations

We found fewer studies than we anticipated, potentially due to the challenges of collecting spatial data in developing countries where most areas do not yet have municipal address systems or subdistrict postal code systems.[15 43] Our search strategy was designed to be inclusive, and we did identify one previous systematic review of spatially targeted intervention for TB.[44] However, for data synthesis we included only primary manuscript sources and ensured that studies included in the previous, more narrowly focused systematic review were also included here. Nevertheless, in recent years technologies for obtaining spatial data have improved, become cheaper and hence more widely available; consequently, the inclusion of spatial data within epidemiological and surveillance studies as a tool to identify disease hotspots has

become more common.[42] In theory, this should enable disease control programmes to collect spatial data for identifying and targeting hotspots.[42 43] Thus, in future, prioritising complete mapping of low income countries, with introduction of effective residential identification systems should be a major priority. This would allow some of the methods for geolocation of notified cases in the studies in this review, which used manual and labour intensive approaches that also had high potential for imprecision, to be replaced with more efficient approaches.[33]

Investment in national geospatial and surveillance tools that allow rapid, accurate and scalable geolocation of cases from within health facilities in settings that lack postal codes would greatly facilitate spatial mapping of areas of interest.[45] Such tools would enable the identification of hotspots at finer scale and limit the use of arbitrary blocks.[34] One important concern about targeting interventions on the basis of case-notifications alone, however, would be the potential for ascertainment bias to further disadvantage underserved populations from which cases are already under-notified due to limited access to health services.[46] For populations where this is likely, the ideal approach would be to investigate disease burden using prevalence surveys, or developing models that include data on geographical distribution of measures of poverty and health service access that can be then allow routine notification data to be adjusted to account for likely under-ascertainment.[46]

We report a wide variety of interventions in the studies included in this systematic review. For HIV,[29] TB[24 29 31] and leprosy,[25 32 33] studies tended to focus on community screening interventions in suspected geographical hotspots. Malaria studies,[34–36] however, were more focused on complex interventions that combined community screening and MDA, with environmental measures to address the vector, including larviciding of stagnant water bodies and IRS.

Despite the above reservations, the studies that we have identified demonstrate the feasibility of the spatially targeted approach to direct resources towards community-based interventions. For case-finding interventions or community screening, spatially targeted interventions are a compromise between tracing known contacts of infectious cases and ACF interventions across the entire general populations or subpopulation (ie, whole urban slums).[47] One limitation of contact tracing is that it misses out cases that are unknown to the index case, and so will fail to identify cases arising from transmission to casual contacts outside of the household.[48 49] Spatially targeted interventions become increasingly important for diseases that are approaching elimination, since these conditions predispose the disease cases to cluster into disease hotspots.[13–19] Hence, spatially targeted interventions can help the most successful control programmes to systematically identify and address residual disease hotspots.[12 19 32 50] Among all of the studies reviewed here, only one (Bousema *et al*[36]) was assessed as having low risk of bias. This was a malaria intervention that evaluated the impact of a spatially targeted intervention in predefined inner and outer regions of hotspots (evaluation zones). This study design allows the evaluation of how targeting the hotspots benefits the individuals that are in the hotspot as well as the individuals that are in the outer regions of the hotspot. This in turn allows assessment of the impact of the intervention on reducing transmission. For vector borne diseases, including malaria, consideration need to be given to the mobility of the vectors themselves when considering spatially targeted interventions[51] that is, if mosquitoes are able to travel over moderate distances the effect of a hotspot targeted intervention might end up being diluted.

However, this study had several limitations.[36] It was underpowered to detect small changes due to the hotspot targeted intervention. In addition, the intervention was also done in one transmission season which might not have been enough to interrupt the epidemic.[36] The area of study might also not have been suitable for a hotspot targeted intervention because even though malaria was heterogeneous there might have been widespread transmission of malaria going on in the non-hotspot areas; this might have meant that a non-targeted approach would have been more appropriate for this setting.[52]

Overall, our systematic review's limitation was that the studies we identified were heterogeneous, and we were unable to do a meta-analysis as planned. Also, the studies' methodological quality had challenges due to the lack of validation of disease hotspots and the lack of random allocation of clusters to interventions. We also only included manuscripts in English due to resource limitations, and only four diseases were included. Despite these limitations, we were able to show the potential that spatially targeted interventions have towards improving disease epidemiology.

### Recommendations

In general spatially targeted interventions are only justifiable in settings where the epidemic is heterogenous and where the non-hotspot areas cannot sustain sufficient transmission of the disease.[36] The influence of the hotspots on the epidemic of the disease in the surrounding communities also depends on the effective contact rates between the residents of the hotspots and the surrounding areas; in the case of malaria this also depends on how far mosquitoes can travel between the hotspots and the surrounding communities. Careful attention needs also to be given to identifying hotspots to make sure that the hotspots capture the true burden of the disease by making sure that the hotspots are persistent in time and are adjusted for confounding.[51–53]

### CONCLUSION

The rapid increase in cheap, reliable, geolocation technology now provides such a significant tool in the fight to control and eradicate endemic and epidemic infectious diseases that support for rapid development of effective

patient mapping systems should be considered essential for health systems in low income countries. Ideally, mapping should be combined with statistical modelling to identify hotspots with the most pressing need for intervention. In this systematic review we provide some evidence in support of the effectiveness of spatially targeted intervention, but with the major conclusion being that the current body of evidence is weak. There is an urgent need to define optimal methodology, allowing recommendations to be made to support the design of more rigorous studies that allow clear evaluation of likely impact to be made for the major categories of infectious diseases, including vector-borne and those transmitted person-to-person. Recommendations should include strategies for timely identification of hotspots and the critical aspects needed to measure the effect of intervention strategies targeting those hotspots.

**Author affiliations**
[1]Public Health, Malawi-Liverpool-Wellcome Trust Clinical Research Programme, Blantyre, Malawi
[2]Medical Statistics, London School of Hygiene and Tropical Medicine, London, UK
[3]MRC Clinical Trials Unit, University College London, London, UK
[4]Helse Nord Tuberculosis Initiative, University of Malawi College of Medicine, Blantyre, Malawi
[5]School of Public Health, Yale University, New Haven, Connecticut, USA
[6]Department of Clinical Research, London School of Hygiene & Tropical Medicine, London, UK
[7]Department of Clinical Sciences, Liverpool School of Tropical Medicine, Liverpool, UK

**Acknowledgements** We would like to thank the LSHTM library staff for assisting us with the formulation of the search strategy.

**Contributors** MK, PMP, ELC and JRC designed the study. MK, PMP, JRC and MN contributed to the data collection, management and data analysis. All authors contributed to the data interpretation. MK wrote the first draft. All reviewed, contributed to and approved the final draft. All authors read and approved the final manuscript.

**Funding** This work was supported by Overseas Collaborative Site Staff-Studentship included as part of Professor Elizabeth L Corbett's (ELC) Senior Research Fellowship in Clinical Science from Wellcome Trust ("Sustainable Community Action for Lung hEalth (SCALE): a cluster-randomised trial in Blantyre, Malawi." WT200901/Z/16/Z.). PMP is funded by Wellcome Trust (206575).

**Disclaimer** The funders had no role in study design; in the collection, analysis, and interpretation of data; in the writing of the report; and in the decision to submit the paper for publication.

**ORCID iDs**
McEwen Khundi http://orcid.org/0000-0003-2718-7576
James R Carpenter http://orcid.org/0000-0003-3890-6206
Marriott Nliwasa http://orcid.org/0000-0002-3100-5512
Ted Cohen http://orcid.org/0000-0002-8091-7198
Elizabeth L Corbett http://orcid.org/0000-0002-3552-3181
Peter MacPherson http://orcid.org/0000-0002-0329-9613

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
