## [Reviewer comments · BMJ Open]

ARTICLE DETAILS

TITLE (PROVISIONAL)	EFFECTIVENESS OF SPATIALLY-TARGETED INTERVENTIONS FOR CONTROL OF HIV, TUBERCULOSIS, LEPROSY AND MALARIA: A SYSTEMATIC REVIEW
AUTHORS	Khundi, McEwen; Carpenter, James; Nliwasa, Marriott; Cohen, Ted; Corbett, Liz; MacPherson, Peter

VERSION 1 – REVIEW

REVIEWER	Paul-Ebhohimhen , VA NHS Highland, Occupational Health and Wellbeing Service
REVIEW RETURNED	30-Nov-2020

GENERAL COMMENTS	The study describes comparative yields or disease notification rates following targeted screening or active case finding for four infectious diseases; of differing organisms, vector and route of transmission, incubation and latency periods, and incidence and prevalence; in various geo-located 'hot spots' . It is noted that given the significant heterogeneity, a meta-analysis across all four infections may not have been anticipated. To meet the descriptors of a systematic review however, more focused clinical question in terms of PICO should have been formulated and the reporting reflect it. The report appears to be a scoping review. The abstract does use the term 'systematic literature review' as study method; however this needs to be changed. In consideration of the 'scoping versus systematic review' point above, the title needs to be changed. Furthermore, the use of 'spatial' rather than 'geo-located' or 'geographically targeted', more so in a covid-19 era with widespread definitions and understanding of 'spatial' or 'social distancing' interventions for prevention and control, makes for a rather misleading header. In summary, the title should more accurately read along the lines of 'A Scoping review of geographically targeted community interventions for identification of HIV, TB, Leprosy and Malaria', and the reporting revised to reflect the various summary points above. More minor points relate to editing, like to ensure the first instance of any abbreviation is clearly outlined, which was not done for Active Case Finding (ACF).
---

REVIEWER	Adewo, Debebe Shaweno The University of Sheffield, HEDS
REVIEW RETURNED	30-Jan-2021

GENERAL COMMENTS	This is an interesting and well-written paper appraising evidence on spatially-targeted public health interventions directed towards major infectious
--

diseases. They conducted a systematic review of evidence from relevant databases using keywords generated by involving different experts. Based on the existing studies that were included in this review, it was not possible to draw a solid conclusion on the effectiveness or lack of effect of spatially targeted intervention as the studies were limited methodologically (data - the underlying data used in those studies to define spatial hotspots might not reflect the true underlying disease burden and design—the studies do not provide proper control (except one)).

The study has numerous strengths: 1. using different quality assessment tools based on the design of the study, 2. discussing main limitations, 3. Indicating future research needs. However, I believe addressing the following few points before it is published would help the readers

Major comments

1. Of the included studies, I found one by Bousema et al in western Keniya evaluating the impact of spatially targeted malaria interventions quite interesting. Although this study was strong in design (randomization and proper comparator), the lack of effect of spatially targeted malaria interventions should be interpreted with great caution and that should be emphasized in this report. I really don't feel comfortable when this study is cited as evidence of the lack of effect of spatially targeted interventions. The main source of concern is that those SaTscan identified clusters were not validated and there is no guarantee that they were true malaria clusters. Content with this assertion, had other cluster detection methods (e.g. Getis Ord, LISA, model-based geostatistics, etc) been used, it is likely that some of the clusters detected in this study might not be detected or vice versa). In addition, given that SaTScan identified 27 clusters (many of which I guess are secondary clusters), there is a potential that some of them might not be true clusters as there were no adjustments for covariates, and also the clusters depend on the user-specified parameters including the radius of the circular window. A recent review also has indicated that there was a considerable proportion (38.8%) of high-burden households (PCR prevalence >70%) outside hotspot boundaries (in Bousema et al study) and were probably able to sustain transmission. (See Figure 1 caption in - <https://www.sciencedirect.com/science/article/pii/S1471492219301941#bb0360>). Also, we cannot be very sure, if the interventions within those clusters were optimum in duration and coverage to interrupt transmission. So I would love to see more descriptions in this manuscript focusing on the potential limitations in that particular article that might explain the lack of effect.

2. Although, spatially targeted interventions were defined from the outset as community interventions that targeted hotspots of disease (increased prevalence/incidence) (page 7, line 177-178), some of the included studies don't seem to quite fulfill this criterion. For example, the Fatima study screened people within 50 meters of an index case, but the study does not give us information how that constitutes a hotspot. A brief explanation might help

Minor comments

1. Hard to follow paragraph on pages 9 to 10: lines 257- 261

VERSION 1 – AUTHOR RESPONSE

Reviewer: 1

Dr. VA Paul-Ebhohimhen , NHS Highland

Comments to the Author:

The study describes comparative yields or disease notification rates following targeted screening or active case finding for four infectious diseases; of differing organisms, vector and route of transmission, incubation and latency periods, and incidence and prevalence; in various geo-located 'hot spots' .

It is noted that given the significant heterogeneity, a meta-analysis across all four infections may not have been anticipated. To meet the descriptors of a systematic review however, more focused clinical question in terms of PICO should have been formulated and the reporting reflect it. The report appears to be a scoping review. The abstract does use the term 'systematic literature review' as study method; however this needs to be changed. In consideration of the 'scoping versus systematic review' point above, the title needs to be changed.

Thank you for these helpful comments. After careful reflection and after taking into account the Editor's guidance, we have decided to retain it as a systematic review.

Furthermore, the use of 'spatial' rather than 'geo-located' or 'geographically targeted', more so in a covid-19 era with widespread definitions and understanding of 'spatial' or 'social distancing' interventions for prevention and control, makes for a rather misleading header.

Thank you for this suggestion, "spatially targeted" is a widely used term and commonly understood in the infectious disease epidemiology community. We feel that introducing new terms might confuse the reader.

In summary, the title should more accurately read along the lines of 'A Scoping review of geographically targeted community interventions for identification of HIV, TB, Leprosy and Malaria', and the reporting revised to reflect the various summary points above.

Thank you for this comment, after reflection and guidance from the Editor above we have decided to retain the title as it is.

More minor points relate to editing, like to ensure the first instance of any abbreviation is clearly outlined, which was not done for Active Case Finding (ACF).

Thank you we have fixed this abbreviation and have carefully checked other abbreviations in the manuscript, to add full words before the first use of an abbreviation.

Line 269 to 270 page 10.

"... concentration of cases were identified as hotspots. Four active case finding (ACF) campaigns were conducted in these hotspots between March and September 2005. Study team members went door..."

Reviewer: 2

Dr. Debebe Shaweno Adewo, The University of Sheffield

Comments to the Author:

Summary

This is an interesting and well-written paper appraising evidence on spatially-targeted public health interventions directed towards major infectious diseases. They conducted a systematic review of evidence from relevant databases using keywords generated by involving different experts. Based on the existing studies that were included in this review, it was not possible to draw a solid

conclusion on the effectiveness or lack of effect of spatially targeted intervention as the studies were limited methodologically (data - the underlying data used in those studies to define spatial hotspots might not reflect the true underlying disease burden and design—the studies do not provide proper control (except one)).

The study has numerous strengths: 1. using different quality assessment tools based on the design of the study, 2. discussing main limitations, 3. Indicating future research needs. However, I believe addressing the following few points before it is published would help the readers
Thank you for your detailed encouraging comments. We have responded to the detailed feedback below:

Major comments

1. Of the included studies, I found one by Bousema et al in western Kenya evaluating the impact of spatially targeted malaria interventions quite interesting. Although this study was strong in design (randomization and proper comparator), the lack of effect of spatially targeted malaria interventions should be interpreted with great caution and that should be emphasized in this report. I really don't feel comfortable when this study is cited as evidence of the lack of effect of spatially targeted interventions. The main source of concern is that those SaTscan identified clusters were not validated and there is no guarantee that they were true malaria clusters. Content with this assertion, had other cluster detection methods (e.g. Getis Ord, LISA, model-based geostatistics, etc) been used, it is likely that some of the clusters detected in this study might not be detected or vice versa). In addition, given that SaTScan identified 27 clusters (many of which I guess are secondary clusters), there is a potential that some of them might not be true clusters as there were no adjustments for covariates, and also the clusters depend on the user-specified parameters including the radius of the circular window. A recent review also has indicated that there was a considerable proportion (38.8%) of high-burden households (PCR prevalence >70%) outside hotspot boundaries (in Bousema et al study) and were probably able to sustain transmission. (See Figure 1 caption in - <https://www.sciencedirect.com/science/article/pii/S1471492219301941#bb0360>). Also, we cannot be very sure, if the interventions within those clusters were optimum in duration and coverage to interrupt transmission. So I would love to see more descriptions in this manuscript focusing on the potential limitations in that particular article that might explain the lack of effect.

Thanks so much for seeking this clarification. We have improved the description of study limitations by adding the details below to the paper lines 416-437 on pages 19 to 20:

“However, this study had several limitations (36). It was underpowered to detect small changes due to the hotspot targeted intervention. In addition, the intervention was also done in one transmission season which might not have been enough to interrupt the epidemic (36). The area of study might also not have been suitable for a hotspot targeted intervention because even though malaria was heterogeneous there might have been widespread transmission of malaria going on in the non-hotspot areas; this might have meant that a non-targeted approach would have been more appropriate for this setting (51).

We have additionally added text to discuss the overall limitations of this review:

“Overall, our systematic review's limitation was that the studies we identified were heterogeneous, and we were unable to do a meta-analysis as planned. Also, the studies' methodological quality had challenges due to the lack of validation of disease hotspots and the lack of random allocation of clusters to interventions. We also only included manuscripts in English due to resource limitations, and only four diseases were included. Despite these limitations, we were able to show the potential that spatially targeted interventions have for improving disease epidemiology.”

Finally, we have updated the Recommendations paragraph to ensure these limitations are highlighted as important areas for improvement in future studies.

“Recommendations

In general, spatially-targeted interventions are only justifiable in settings where the epidemic is heterogeneous and secondly where the non-hotspot areas cannot sustain sufficient transmission of the disease (36). The influence of the hotspots on the epidemic of the disease in the surrounding communities also depends on the effective contact rates between the residents of the hotspots and the surrounding areas; in the case of malaria this also depends on how far mosquitoes can travel between the hotspots and the surrounding communities. Careful attention needs also to be given to identifying hotspots to make sure that the hotspots capture the true burden of the disease by making sure that the hotspots are persistent in time and are adjusted for confounding (50–52).”

2. Although, spatially targeted interventions were defined from the outset as community interventions that targeted hotspots of disease (increased prevalence/incidence) (page 7, line 177-178), some of the included studies don't seem to quite fulfill this criterion. For example, the Fatima study screened people within 50 meters of an index case, but the study does not give us information how that constitutes a hotspot. A brief explanation might help

Thank you for this helpful comment. The studies that were included had different definitions for hotspots and we took the definitions based on each study. But we agree that the mentioned study might not have fully fit the definition of a spatially targeted intervention as defined in this review. We have removed it from the included studies, we have also removed one malaria study that had a similar definition of hotspots, screening of individuals in the radius around the household of an index case. After careful review of these studies, we feel they fall under contact tracing rather than spatially targeted interventions.

We have added text to clarify the hotspot definition by noting that contact tracing studies based on geolocated cases were not sufficient for inclusion in this study Lines 177-182, page 7.

“We defined spatially-targeted interventions as community interventions that targeted hotspots of disease. Hotspots were defined as sub-district geospatially-defined areas that had a high number of incident or prevalent cases of the infection or disease compared to surrounding areas, studies that based their hotspots as around diagnosed index cases were not included. In practice, as definitions varied considerably between studies, we extracted and compared hotspot definitions between studies.”

In addition we have additionally updated the Supplemental Table to describe these reasons for exclusion.

Minor comments

1. Hard to follow paragraph on pages 9 to 10: lines 257- 261

We have revised the highlighted lines 251 -263 on pages 9 to 10.

“In Cegielski et al. (31), notified TB cases in Smith County, Texas USA between 1985 to 1995 (N=128) and all notified LTBI cases from 1993 to 1995 (N=311) were geocoded to their households. The geocoded cases were loaded into geographical information systems (GIS) software to produce a point map of both active TB and LTBI cases. The two densest neighbourhoods of TB and LTBI cases were identified visually from the map. In 1996, study field workers went door to door in these neighbourhoods and offered tuberculin skin testing (TST). 1236/2291/2258 had LTBI testing and received results, of whom 1236 (95.7%) had results read and 229/1236 (18.5%) were TST positive and 147 received treatment. To assess the intervention, the notified TB cases from 1996 to 2006 in Smith County were mapped to do a before and after intervention comparison. The TB notification rates in the targeted hotspots declined from 39.6 per 100,000 people per year (95% CI:30.4-48.8)

from 1985 to 1995 to zero from 1996 to 2006 ($p < 0.001$). While for the entire Smith County the TB notification rates reduced from 8.1 per 100,000 people per year (95% CI: 5.2-11.0) from 1985 to 1995 to 3.7 per 100,000 people per year (95% CI: 1.2-6.1) from 1996 to 2006 ($p < 0.001$).”

VERSION 2 – REVIEW

REVIEWER	Paul-Ebhohimhen , VA NHS Highland, Occupational Health and Wellbeing Service
REVIEW RETURNED	29-Apr-2021

GENERAL COMMENTS	Comments to the initial manuscript have been addressed in the main. One final recommendation is to provide the rationale for excluding systematic reviews relevant to either the diseases or interventions, given the following considerations. Firstly, the position of systematic reviews in the hierarchy of evidence; secondly their advantage (where available) in being a source of relevant primary studies during the search; and that in the process, further justification for the systematic review is made when confirmed that no prior similar study exists.
---

REVIEWER	Adewo, Debebe Shaweno The University of Sheffield, HEDS
REVIEW RETURNED	19-Apr-2021

GENERAL COMMENTS	Thanks for taking time to incorporate the suggestions. The authors have addressed all my suggestions and I have no further comments.
--

VERSION 2 – AUTHOR RESPONSE

Reviewer: 1

Dr. VA Paul-Ebhohimhen , NHS Highland

Comments to the Author:

Comments to the initial manuscript have been addressed in the main. One final recommendation is to provide the rationale for excluding systematic reviews relevant to either the diseases or interventions, given the following considerations. Firstly, the position of systematic reviews in the hierarchy of evidence; secondly their advantage (where available) in being a source of relevant primary studies during the search; and that in the process, further justification for the systematic review is made when confirmed that no prior similar study exists.

Thank you so much for your feedback. Please find our response below.

Our systematic review looked at a very specific question: The effect of spatially targeted interventions. Our search strategy was designed to be inclusive, and we did identify one previous systematic review of spatially targeted intervention for TB (1). However, for data synthesis we included only primary manuscript sources and ensured that studies included in the previous, more narrowly focused systematic review were also included here.

We have added this text to the discussion page 18 (line 370 to 371)

“Our search strategy was designed to be inclusive, and we did identify one previous systematic review of spatially targeted intervention for TB (44). However, for data synthesis we included only primary manuscript sources and ensured that studies included in the previous, more narrowly focused systematic review were also included here.”